# The Influence of Geopolitical Risk on International Direct Investment and Its Countermeasures

**Miaozhi Yu and Na Wang ***

School of Economics, Zhejiang University of Technology, Hangzhou 310023, China

* Correspondence: wangna531@outlook.com

**Abstract:** Recent years have seen frequent geopolitical conflicts and the world economy has fallen into a recession. In order to explore how wars, terrorist attacks and international tensions affect foreign direct investment (FDI), this paper uses the fixed-effect model to investigate the impact of geopolitical risks on FDI flows in 41 countries during 2003–2020 from the perspective of market seeking, natural resource seeking and strategic resource seeking. The results show that, on the whole, geopolitical risks can significantly inhibit the inflow of foreign direct investment and hinder the development of domestic economy. The market size, natural resources and science and technology of the host country are important factors to attract foreign investment. Trade dependence has a moderating effect on the negative impact of geopolitical risks. Countries that depend on international trade may eliminate geopolitical frictions through economic cooperation. The impact of geopolitical risk is heterogeneous in countries with different levels of economic development. The impact of geopolitical risk on foreign direct investment in developed economies is not significant.

**Keywords:** geopolitical risks; foreign direct investment; moderating effect

## 1. Introduction

In recent years, geopolitical events such as Sino-US trade friction, Brexit, the Russia-Ukraine conflict and China-India tensions have occurred frequently. The Nord Stream gas pipeline explosion caused by the Russia-Ukraine war and the explosion of the Crimean bridge will further escalate the tension of global political situation. These shocks have fueled growing nationalism and populism, created uncertainty in the global political economy, weakened the world's ability to respond to the crisis of the COVID-19 pandemic, generated new geopolitical tensions, exposed the fragility of the international relations system, and pushed the world economy into recession. According to the World Investment Report released by UNCTAD in 2021 [1], global international direct investment flows in 2020 dropped from US $1.5 trillion in 2019 to US $1.0 trillion, a decline of about 35% and nearly 20% lower than that of US $1.2 trillion in 2009 during the financial crisis (United Nations Conference on Trade and Development, 2021). As the world is undergoing profound changes unseen in a century, global value chains have suffered significant impacts and transnational investment faces increasing uncertainties in the external environment. According to the World Economic Forum's annual Global Risks Report [2], geopolitical risks have been among the top three major global risks in terms of impact for five years in a row. Failure to develop strategies to address them in a timely manner could reverse years of progress in reducing poverty and inequality and further weaken social cohesion and friendly cooperation among countries (World Economic Forum, 2021).

Therefore, it is increasingly important to study geopolitical risks from the perspective of economics. Furthermore, exploring the impact of geopolitical risks on international capital flows is also of great theoretical and practical significance for countries to accurately

identify risks and formulate corresponding policies to attract foreign capital to promote economic development. The study of the moderating variable and national heterogeneity on the impact of geopolitical risks on international investment is helpful for countries to refer to in order to formulate corresponding policies.

A general review of the existing literature shows that geopolitics, as an interdisciplinary subject of geography, political economy and international relations, has been studied and discussed by many scholars, but there is no unified definition of this term. Generally speaking, the broad definition of geopolitical risk includes three aspects: political (war, social unrest, religious conflicts, etc.), economic (trade friction, trade protectionism, anti-globalization, etc.) and natural (earthquake, tsunami, drought, etc.) [3,4]. The narrow definition of geopolitical risk only includes the threat brought by events related to military conflicts, terrorist acts and international tensions [5] and thus some scholars see this as part of the policy uncertainty. Therefore, the measurement of this risk also has different consequences and economic implications depending on the definition, but there is a broad consensus that geopolitical risks, both broad and narrow, will have adverse impacts on the world economic environment. Many scholars have found that geopolitical risks will not only worsen the international relations system [6–8], but also cause fluctuations and instability of macroeconomic data such as global oil prices [9], stock market returns [10,11], commodity prices [12] and policy uncertainties [5]. This accompanying macroeconomic fluctuation and policy uncertainty will again aggravate the potential risks faced by international economic activities [13,14]. In addition, geopolitical risks will inhibit international trade flows [15] and endanger the long-term and stable development of banking [16], energy, insurance [17,18] and tourism industries [19] around the world. From an investment standpoint, Khattab [20] shows that the social risks of the host country and the political threats of the surrounding countries have a significant negative impact on foreign investment. Li and Vashchilko [21] provide consistent evidence that military conflicts can reduce bilateral investment between high-income countries and low-income countries. Anh-Tuan and Thao [22] demonstrate that geopolitical risk is a crucial macrolevel shock influencing corporate investment. Caldara and Iacoviello [5] also simply verified the negative correlation between geopolitical risk and international capital flows.

However, there are relatively few studies focusing on the impact of geopolitical risk and foreign direct investment in the literature. Most of the existing literature related to geopolitical risk is from the perspective of international relations and, because the authors have different definitions of geopolitical risk, they have also obtained different results. In addition, prior studies only document the impact of geopolitical related risks on economic activities, but rarely do they study the economic factors that can adjust the impact, and thus explore how to reduce the inhibition of this impact on economic growth in policy. Hence, compared with the existing literature, this paper only focuses on the impact of geopolitical risk in a narrow sense on an economy's ability to attract foreign direct investment (FDI), and will make contributions in the following two aspects: (1) explore the moderating effect of the host country's trade dependence on this inhibitory effect, and provide empirical reference value for policy makers; and (2) analyze the heterogeneity of economies with different levels of development affected by geopolitical risks to enrich the academic experience in this field.

The second part of this paper is a theoretical analysis and research hypothesis from the theoretical level of geopolitical risk on the impact of international direct investment analysis, and thus puts forward three hypotheses. The third part, data and methodology, mainly expounds the construction of the empirical model, data sources and the selection of explained variables, core explanatory variables and control variables. The fourth part is empirical results. This paper uses the fixed-effect model to conduct regression analysis for three hypotheses, including baseline regression, robustness test, moderating effect and heterogeneity analysis. The fifth part is discussion and conclusions. It puts forward suggestions based on the theoretical analysis and empirical regression results and summarizes the research results of this paper.

## 2. Theoretical Basis and Research Assumptions

According to the theory of comparative advantage, reducing trade barriers can promote world economic growth through specialization. However, with the development of global trade liberalization, capital is gradually transferred from developed economies to emerging economies, which inevitably damages the interests of developed economies [3]. Therefore, in order to retain the capital lost due to globalization and promote domestic employment, some developed countries began to increase trade barriers, which led to a surge of trade protectionism, which undoubtedly increased the risk of global economic activities, posed a threat to world economic growth, and intensified tensions in international relations. The risks arising from geopolitical events will inhibit domestic economic growth by reducing the flow of international trade and capital [23], thus further straining international relations and promoting the shift of the global order from rules-based to power-based, forming a vicious circle.

According to the traditional international investment theory, the uncertainty of the host country's economic environment is one of the main sources of potential economic costs. Faced with such risks, transnational corporations will adopt a wait-and-see attitude to postpone outward foreign direct investment (OFDI) or redistribute the investment to countries with lower risks. Therefore, higher political and economic risks will inhibit FDI [24–26]. Moreover, foreign investment is more sensitive to the political environment of the host country than domestic investment due to the limited political and economic information available to foreign investors and limited legal protections [27]. However, some scholars questioned the idea that investors have "risk-off behavior" and found that developing countries such as China have the characteristics of institutional risk appetite, that is, they are more inclined to invest in countries with political instability [28]. Some scholars have given explanations for this phenomenon. On the one hand, it is because the investment in these countries faces less competition from developed countries; on the other hand, the main body of China's OFDI is state-owned enterprises, and political motives such as enhancing diplomatic relations between the two countries have become the main factors affecting investment decisions [29]. However, considering the economic factors such as the level of economic development and natural resource endowment of the host country, the investment preferences of developing countries are different under different investment motivations. The overall investment preferences still show the characteristics of risk aversion, which is in line with the international mainstream investment theory [30,31].

Therefore, we can assume that international capital flows from regions with high geopolitical risk to regions with low geopolitical risk.

**Hypothesis 1.** *Geopolitical risk is negatively correlated with foreign direct investment flows.*

The trade peace theory can be traced back to the 17th century liberal economist Emerick Kreusser, who argued that free trade could reap the same benefits as occupation and conquest. Since then, Adam Smith and David Ricardo have developed and perfected the theory of free trade, arguing that international trade can promote world peace [32]. This theory holds that international trade can inhibit political and military conflicts between countries, and it has wide support [33]. With the increase in national trade openness, the peaceful effect of international trade will also be enhanced. Countries with high trade dependence may wish to establish favorable economic relations, thus reducing military conflicts [34]. While non-liberal scholars have challenged this view, arguing that trade does not necessarily have a significant impact on international peace if politicians do not prioritize the economic costs of actions such as war [35], studies have shown that both wars and acts of terrorism can destabilize a country's economy and thus inhibit economic growth [36,37]. According to the theory of irreversible investment, future political and economic uncertainty will reduce the investment of risk-neutral firms, because rational multinational investors will consider that sunk costs cannot be recovered in the event of adverse market changes [38], and the higher the degree of irreversibility of investment, the greater the re-

straining influence of geopolitical risk on transnational investment [22,39]. In other words, a stable macro-political and economic environment is essential to attract foreign direct investment flows into a country and thus promote its own economic growth [40,41]. Therefore, for policy makers, countries with a high dependence on export trade will pay more attention to political peace and stability in order to encourage the introduction of foreign capital, or formulate corresponding policies to provide security for transnational investors.

So, we can assume that for countries or regions with high trade dependence, geopolitical risks will have less impact on FDI.

**Hypothesis 2.** *The host country's trade dependence has a moderating effect on the influence of geopolitical risks on FDI flows.*

According to the eclectic paradigm theory and investment development path theory represented by British economist Dunning, foreign investment in developed countries is mainly concentrated in technology-intensive industries, while foreign investment in developing countries is mainly concentrated in labor-intensive industries [42,43]. For multinational investors, labor factors of production are more fungible than science and technology, and developed countries have more complete safeguard measures than developing countries in the face of various external shocks. Studies have shown that the higher the per capita real gross national product (GNP), the more foreign direct investment it attracts [44]. Complete infrastructure construction and extensive popularization of higher education are also one of the positive factors to promote the introduction of foreign capital [45].

Therefore, we can assume that in the face of countries with the same geopolitical risks, investors will tend to make transnational investments to countries with higher economic development level. In other words, compared with emerging economies, geopolitical risks in developed economies will have less inhibitory effect on FDI flows.

**Hypothesis 3.** *The impact of geopolitical risks on FDI flows in developed and emerging economies is heterogeneous.*

## 3. Data and Methodology

### 3.1. Model Setting

About baseline regression, limited by the availability of data and by referring to the basic theories of international direct investment and relevant literature, this paper selects the foreign direct investment data of 41 countries and regions (including Argentina, Australia, Belgium, Brazil, Canada, Switzerland, Chile, China, Colombia, Germany, Denmark, Egypt, Spain, Finland, France, the United Kingdom, Hong Kong, Hungary, Indonesia, India, Israel, Italy, Japan, South Korea, Mexico, Malaysia, the Netherlands, Norway, Peru, the Philippines, Poland, Portugal, Russia, Saudi Arabia, Sweden, Thailand, Tunisia, Turkey, Ukraine, United States, and South Africa) from 2003 to 2020. The regression model is as follows:

$$ifdi_{it} = \alpha_0 + \alpha_1 gpr_{it} + \alpha_2' Z_{it} + \lambda_t + u_i + \varepsilon_{it} \tag{1}$$

wherein the explained variable $ifdi_{it}$ represents the annual foreign direct investment flow of the host country $i$ in the year $t$, $\alpha_0$ is the intercept term. The core explanatory variable $gpr_{it}$ represents the annual geopolitical risk index of the host country $i$ in the year $t$, and the sign expectation of the coefficient $\alpha_1$ is negative; the economic implication is that the geopolitical risk of the host country will inhibit the inflow of foreign direct investment. $Z_{it}$ is a series of control variables, including the market size of the host country that drives the inflow of international direct investment, natural resource endowment and macroeconomic data of strategic resources. $\alpha_2$ is a k × 1 parameter vector, which represents the coefficient of these control variables, k is the number of control variables. $\lambda_t$ is time fixed effect, which controls the common time-varying factors affecting the sample countries and regions, such as global macroeconomic shocks or policy changes, including external shocks and indirect effects of policy uncertainties in other countries and regions; $u_i$ is the individual fixed effect

controlling the factors that do not change over time in the sample countries and regions, such as distance, common language, culture, etc. $\varepsilon_{it}$ is the random disturbance term. $i$ represents 18 years from 2003 to 2020, $t$ represents 41 countries and regions.

About the modulating effect, this paper uses panel data of 41 countries from 2005 to 2020 to empirically study the moderating effect of the host country's trade dependence on the inhibition of foreign direct investment by geopolitical risk. The regression model is as follows:

$$ifdi_{it} = \alpha_0 + \alpha_1 gpr_{it} + \alpha_2 gpr_{it} * e_{it} + \alpha_3' Z_{it} + \lambda_t + u_i + \varepsilon_{it} \tag{2}$$

wherein $\alpha_0$ is the intercept term. $\alpha_1$ is the coefficient of the core explanatory variable $gpr_{it}$. The explanatory variable $gpr_{it} * e_{it}$ is the interaction term between the geopolitical risk index and the host country's trade dependence. The economic meaning of the coefficient $\alpha_2$ of the interaction term is the influence of geopolitical risk on FDI inflow under the influence of trade dependence. $\alpha_3$ is the coefficient of these control variables. Other variables were set in accordance with the baseline regression model. In this paper, the percentage of merchandise exports to GDP is used to measure the host country's trade dependence, which is derived from the WDI database of the World Bank.

*3.2. Variable Selection and Data Description*

Explained variable: annual foreign direct investment flows (ifdi) for 41 countries and regions, data from the World Bank WDI database.

Core explanatory variable: Geopolitical Risk Index (gpr). The data comes from the monthly geopolitical risk index published by Caldara and Iacoviello [5], which was averaged over 12 months to obtain its annual index (https://www.matteoiacoviello.com/gpr.htm accessed on 17 September 2022). The index reflects the automatic text search results of the electronic edition of 10 English-language newspapers (Chicago Tribune, the Daily Telegraph, Financial Times, The Globe and Mail, The Guardian, the Los Angeles Times, The New York Times, USA Today, The Wall Street Journal, and The Washington Post) with high circulation. It measures the geopolitical risk of a country or region by calculating the number of articles related to adverse geopolitical events in each month's newspaper as a proportion of the total number of news articles. Empirical results show that the index is not affected by media bias, economic recession and financial crisis.

Control variables: By referring to the traditional international investment theory and location advantage theory, as well as the variable selection of transnational investment motivation by the relevant literature [46–48], this paper selects the following indicators as the proxy variables of investment motivation.

1.  Market seeking motivation, the motivation of foreign direct investment by multinational enterprises in order to maintain the existing international market share or seek the same market competitiveness as local enterprises overseas: Per capita GDP growth rate (lngdpc) was used as the proxy variable to measure the market size of countries and regions, and logarithmic processing was carried out, with data from the WDI database of the World Bank;

2.  Natural resource seeking motivation, the motivation of foreign direct investment by multinational enterprises in search of a stable and cheap supply of natural resources: Natural resource endowments of countries and regions were measured using fuel exports as a percentage of commodity exports (fuel_ex) and ores and metals exports as a percentage of commodity exports (ores_metals_ex), derived from the World Bank WDI database;

3.  Strategic resource seeking motivation, the motivation of foreign direct investment by multinational enterprises to acquire and utilize the advanced science and technology or management experience of the host country: The number of resident and non-resident patent applications (patent_re and patent_non) was used as proxy variables for the strategic resources of countries and regions to attract foreign investment. The data was derived from the WDI database of the World Bank.

Table 1 shows variable selection, acronym and data sources. The descriptive statistical results are shown in Table 2.

**Table 1.** Variable selection.

| | Variables | Acronym | Data Sources |
|---|---|---|---|
| Explained variable | Foreign direct investment flows | ifdi | WDI |
| Core explanatory variable | Geopolitical risk index | gpr | Geopolitical Risk (GPR) Index. Available online: https://www.matteoiacoviello.com/gpr.htm (accessed on 17 September 2022) [49] |
| Control variables | Market seeking motivation — Per capita GDP growth rate | lngdpc | WDI |
| | Natural resource seeking motivation — Fuel exports as a percentage of commodity exports | fuel_ex | WDI |
| | Natural resource seeking motivation — Ores and metals exports as a percentage of commodity exports | ores_metals_ex | WDI |
| | Strategic resource seeking motivation — Number of resident patent applications | patent_re | WDI |
| | Strategic resource seeking motivation — Number of non-resident patent applications | patent_non | WDI |

**Table 2.** Descriptive statistics.

| Variables | Obs. | Mean | St. Dev | Min | Max |
|---|---|---|---|---|---|
| ifdi | 738 | 369.900 | 748.200 | −3444.000 | 7338.000 |
| gpr | 738 | 21.860 | 41.120 | 0.427 | 435.000 |
| lngdpc | 738 | 11.660 | 2.127 | 8.177 | 17.890 |
| fuel_ex | 736 | 13.850 | 18.950 | 0.026 | 91.370 |
| ores_metals_ex | 737 | 7.012 | 11.370 | 0.122 | 64.450 |
| patent_re | 726 | 36,489.000 | 135,549.000 | 27.000 | $1.394 \times 10^6$ |
| patent_non | 726 | 17,141.000 | 44,523.000 | 18.000 | 336,340.000 |

Note: This table shows the descriptive statistics. The definitions of the variables are provided in Table 1.

## 4. Empirical Results

### 4.1. Basic Regression

Using a Hausman test to see if all explanatory variables are not associated with individual effects, the results in Table 3 show that the null hypothesis is strongly rejected. Therefore, the fixed-effect model should be used.

Five sets of models are used to examine the impact of geopolitical risks on foreign direct investment flows in 41 countries and regions from 2003 to 2020. Table 3 shows the full sample benchmark regression results of the clustering fixed-effect model. No control variables were added to Model 1, and only the influence of the core explanatory variable, namely geopolitical risk, was considered. Model 2 controls the relative value of per capita GDP of the host country from the perspective of market seeking motivation. Model 3 uses the natural resource endowment of the host country as a control variable from the perspective of natural resource seeking motivation. Model 4 uses the patent registration data of the host country as the control variable from the perspective of strategic resource seeking motivation. In Model 5, all control variables were added to examine the stability of regression results.

**Table 3.** Baseline regression of the full sample.

| Variables | (1) | (2) | (3) | (4) | (5) |
|---|---|---|---|---|---|
| gpr | −3.026 *** | −3.189 *** | −2.669 *** | −1.711 ** | −1.577 ** |
| | (−4.593) | (−5.278) | (−4.106) | (−2.258) | (−2.230) |
| lngdpc | | 112.659 * | | | 61.248 ** |
| | | (1.847) | | | (2.038) |
| fuel_ex | | | 5.697 ** | | 4.731 * |
| | | | (2.303) | | (1.730) |
| ores_metals_ex | | | 12.887 ** | | 12.024 ** |
| | | | (2.021) | | (2.137) |
| patent_re | | | | 0.000 * | 0.000 * |
| | | | | (1.880) | (1.714) |
| patent_non | | | | 0.009 *** | 0.009 *** |
| | | | | (4.268) | (4.721) |
| Constant | 436.084 *** | −874.438 | 259.653 *** | 242.644 *** | −613.206 |
| | (30.275) | (−1.229) | (3.787) | (9.651) | (−1.677) |
| Observations | 738 | 738 | 736 | 726 | 725 |
| R-squared | 0.007 | 0.015 | 0.011 | 0.048 | 0.053 |
| AIC | 11,273.6 | 11,269.8 | 11,246.1 | 11,074.3 | 11,062.5 |
| BIC | 11,278.2 | 11,279.0 | 11,259.9 | 11,098.1 | 11,080.0 |
| Number of countries | 41 | 41 | 41 | 41 | 41 |
| | Hausman | | | 59.01 | |
| | *p*-value | | | $2.38 \times 10^{-10}$ | |

Note: This table shows the results for testing the effect of geopolitical risk on foreign direct investment. The definitions of the variables are provided in Table 1. Country and year fixed effects are included in the models. The coefficient estimates and *t*-statistics are reported based on robust standard errors clustered by country. *** $p < 0.01$, ** $p < 0.05$, * $p < 0.1$; in parentheses are the *t*-statistics of the corresponding coefficients.

The regression results show that the coefficient of geopolitical risk gpr in the five groups of models is significantly negative, and passes the test at the significance level of 1%; that is, the geopolitical risk of the host country has a significant negative impact on the inflow of foreign direct investment. Therefore, hypothesis 1 is valid, which conforms to the behavior characteristic of investors in the international investment theory of "seeking advantages and avoiding disadvantages". The coefficients of all control variables in Model 2–5 are significantly positive, indicating that market seeking motivation, natural resource seeking motivation and strategic resource seeking motivation are important factors to promote FDI. In other words, the higher the economic development level of the host country, the richer the natural resources and the more developed the science and technology, the easier it is to attract the eyes of transnational investors, so as to introduce foreign capital to promote the development of the national economy, forming a virtuous cycle which conforms to the location advantage theory. In addition, the coefficient of the core explanatory variable in Models 3–5 is significantly smaller than 3.026 in Model 1, indicating that when the host country is rich in natural and strategic resources, the negative impact of geopolitical risks on FDI inflows will be weakened to some extent. This may be because multinational investors need to consider cost and return comprehensively when making investment decisions. If the expected return is greater than the opportunity cost, the risk aversion caused by the rising cost can be offset to some extent.

### 4.2. Robustness Checks

Considering the international status and location advantages of the United States and the political particularity of Hong Kong, China, these two economies are excluded from the sample selection in this section, and thus change the indicators used by each variable. Since this paper studies geopolitical risks in a narrow sense, the external shock index of war and other actions (ex_conflict), and the percentage of military expenditure to GDP (military_gdp) published by ICRG, are used to replace the core explanatory variables with

data from ICRG country risk reports over the years and from the WDI database of the World Bank. The control variables are replaced with GDP growth rates (gdpg), coal rents (coalr), mineral rents (mineral_rents) and commodity exports to high-income economies as a percentage of total commodity exports (ex_high), all from the World Bank WDI database. In addition, limited by the availability of data, the sample years were changed to 2003–2016. Table 4 shows the new variable selection, acronym and data sources. Clustering fixed effect regression was conducted on the new sample data again, and the results were shown in Table 5.

**Table 4.** Variable selection for robustness checks.

| | **Variables** | | **Acronym** | **Data Sources** |
|---|---|---|---|---|
| Explained variable | | Foreign direct investment flows | ifdi | WDI |
| Core explanatory variable | | External shock index of war and other actions | ex_conflict | ICRG |
| | | Percentage of military expenditure to GDP | military_gdp | ICRG |
| Control variables | Market seeking motivation | GDP growth rates | gdpg | WDI |
| | Natural resource seeking motivation | Coal rents | coalr | WDI |
| | | Mineral rents | mineral_rents | WDI |
| | Strategic resource seeking motivation | Commodity exports to high-income economies as a percentage of total commodity exports | ex_high | WDI |

**Table 5.** Robustness test.

| Variables | (1) | (2) | (3) | (4) | (5) |
|---|---|---|---|---|---|
| ex_conflict | −98.420 * | −109.460 ** | −101.917 * | −84.336 * | −97.508 * |
| | (−1.931) | (−2.116) | (−1.927) | (−1.702) | (−1.890) |
| military_gdp | −98.914 * | −88.727 * | −95.058 * | −81.121 * | −57.449 |
| | (−1.887) | (−1.801) | (−1.893) | (−1.696) | (−1.432) |
| gdpg | | 10.627 | | | 14.251 * |
| | | (1.447) | | | (1.774) |
| coalr | | | 38.069 ** | | 40.228 *** |
| | | | (2.446) | | (3.126) |
| mineral_rents | | | 9.506 | | 5.066 |
| | | | (1.339) | | (0.946) |
| ex_high | | | | −6.519 | −8.899 * |
| | | | | (−1.510) | (−1.975) |
| Constant | 1492.289 ** | 1548.710 ** | 1499.028 ** | 1790.507 *** | 1985.655 *** |
| | (2.583) | (2.664) | (2.542) | (2.795) | (2.928) |
| Observations | 546 | 546 | 546 | 546 | 546 |
| R-squared | 0.022 | 0.027 | 0.025 | 0.027 | 0.038 |
| AIC | 8042.8 | 8042.0 | 8045.3 | 8041.8 | 8041.6 |
| BIC | 8051.4 | 8054.9 | 8062.5 | 8054.7 | 8047.5 |
| Number of countries | 39 | 39 | 39 | 39 | 39 |

Note: This table shows the results of robustness test of the impact of geopolitical risk on foreign direct investment. The definitions of the variables are provided in Table 4. Country and year fixed effects are included in the models. The coefficient estimates and *t*-statistics are reported based on robust standard errors clustered by country. *** $p < 0.01$, ** $p < 0.05$, * $p < 0.1$; in parentheses are the *t*-statistics of the corresponding coefficients.

According to the regression results of Models 1–5 in Table 5, the coefficients of the two core explanatory variables, the external impact index (ex_conflict) and the percentage of military expenditure in GDP (military_gdp) are significantly negative at the 1% level. This result still shows that the risks brought by geopolitical events such as war, terrorist attacks and tensions in international relations will significantly reduce the inflow of FDI, which is consistent with the results obtained in the previous study, which proves that the conclusions of this paper are robust to some extent.

### 4.3. Moderating Effect

In order to analyze the role of the host country's trade dependence in improving the adverse impact of geopolitical risk on foreign direct investment, this paper uses the percentage of commodity exports to GDP to measure the host country's trade dependence, and takes the interaction term between the geopolitical risk index and trade dependence as the core explanatory variable to carry out fixed effect regression again. Table 6 shows the regression results of the moderating effect.

**Table 6.** Regression results of moderating effect.

| Variables | (1) | (2) | (3) | (4) | (5) |
|---|---|---|---|---|---|
| gpr | 9.421 ** | 9.225 * | 9.911 ** | 8.099 | 8.325 |
| | (2.168) | (2.004) | (2.260) | (1.610) | (1.617) |
| gpr*e | −0.261 * | −0.260 * | −0.266 * | −0.247 * | −0.249 * |
| | (−1.891) | (−1.881) | (−1.870) | (−1.709) | (−1.706) |
| lngdpc | | 25.366 | | | 23.020 |
| | | (0.498) | | | (0.504) |
| fuel_ex | | | 6.055 ** | | 4.295 |
| | | | (2.591) | | (1.579) |
| ores_metals_ex | | | 7.858 | | 9.607 |
| | | | (0.961) | | (1.311) |
| patent_re | | | | −0.000 | −0.000 |
| | | | | (−0.871) | (−0.921) |
| patent_non | | | | 0.008 *** | 0.008 *** |
| | | | | (3.261) | (3.390) |
| Constant | 355.294 *** | 61.263 | 207.163 ** | 239.845 *** | −155.899 |
| | (8.127) | (0.104) | (2.481) | (6.876) | (−0.283) |
| Observations | 656 | 656 | 654 | 646 | 645 |
| R-squared | 0.012 | 0.012 | 0.014 | 0.026 | 0.027 |
| AIC | 10,049.4 | 10,051.2 | 10,023.1 | 9899.9 | 9890.6 |
| BIC | 10,058.4 | 10,064.7 | 10,041.1 | 9927.8 | 9921.9 |
| Number of countries | 41 | 41 | 41 | 41 | 41 |

Note: This table shows the results for testing the moderating effect of trade dependence on the impact of geopolitical risk on foreign direct investment. gpr*e is the interaction term between the geopolitical risk index and the host country's trade dependence. Other definitions of the variables are provided in Table 1. Country and year fixed effects are included in the models. The coefficient estimates and *t*-statistics are reported based on robust standard errors clustered by country. *** $p < 0.01$, ** $p < 0.05$, * $p < 0.1$; in parentheses are the *t*-statistics of the corresponding coefficients.

The regression results in Table 6 show that the coefficient of the interaction term between the geopolitical risk index and trade dependence is still significantly negative at the level of 1%, indicating that the geopolitical risk based on the trade peace theory will still significantly inhibit the inflow of foreign direct investment. However, by comparing the regression results of Tables 3 and 6, it can be seen that the coefficients of core explanatory variables in the five groups of models all become significantly smaller, while the coefficients of proxy variables of the three investment motivations do not change significantly. It can be seen that the host country's trade dependence has a significant moderating effect on the negative impact of geopolitical risks on FDI flows, so hypothesis 2 is valid.

The underlying reasons for this phenomenon may be as follows: First, countries that depend on international trade may have developed better laws and regulations to guarantee international economic activities. Second, these countries are more willing to sign trade and investment agreements with other countries to ensure the stable development of their own international trade and investment and make international investment activities more convenient. Third, it is easier for these countries to maintain peaceful and friendly international relations with countries that have established economic exchanges and eliminate geopolitical conflicts through economic exchanges.

### 4.4. Heterogeneity Analysis

In order to investigate whether the negative impact of geopolitical risks on FDI inflow of economies with different levels of economic development is heterogeneous, the sample countries and regions were divided into 22 developed economies (including Australia, Belgium, Canada, Switzerland, Germany, Denmark, Spain, Finland, France, the United Kingdom, Hong Kong, Hungary, Israel, Italy, Japan, South Korea, the Netherlands, Norway, Poland, Portugal, Sweden and the United States) and 19 emerging economies (including Argentina, Brazil, Chile, China, Colombia, Egypt, Indonesia, India, Mexico, Malaysia, Peru, the Philippines, Russia, Saudi Arabia, Thailand, Tunisia, Turkey, Ukraine and South Africa) according to the announcements and data of the United Nations, the World Bank and the International Monetary Fund. The regression results are shown in Table 7. Model 1 is developed economy and model 2 is emerging economy.

**Table 7.** Subsample regression results.

| Variables | Developed Economy | Emerging Economy |
|---|---|---|
| gpr | −0.790 | −4.641 ** |
| | (−1.444) | (−2.589) |
| lngdpc | −51.577 | 33.571 *** |
| | (−0.173) | (3.756) |
| fuel_ex | 5.430 | 2.115 |
| | (0.418) | (1.346) |
| ores_metals_ex | 27.281 * | 5.702 |
| | (1.730) | (1.406) |
| patent_re | 0.001 | −0.001 *** |
| | (0.198) | (−3.084) |
| patent_non | 0.008 *** | 0.031 *** |
| | (4.140) | (5.536) |
| Constant | 754.517 | −527.734 *** |
| | (0.223) | (−3.052) |
| Observations | 390 | 335 |
| R-squared | 0.029 | 0.603 |
| AIC | 6181.6 | 4172.2 |
| BIC | 6205.4 | 4195.1 |
| Number of countries | 22 | 19 |

Note: This table shows the results for testing the impact of geopolitical risk on foreign direct investment of economies with different levels of economic development. The definitions of the variables are provided in Table 1. Country and year fixed effects are included in the models. The coefficient estimates and *t*-statistics are reported based on robust standard errors clustered by country. *** $p < 0.01$, ** $p < 0.05$, * $p < 0.1$; in parentheses are the *t*-statistics of the corresponding coefficients.

The regression results of models 1 and 2 in Table 7 clearly show that for developed economies, geopolitical risks have no significant impact on foreign direct investment inflows, while geopolitical risks in emerging economies significantly inhibit foreign direct investment inflows. Hypothesis 3 is valid.

The impact of geopolitical risks on economies at different levels of development may be different for three reasons: First, cross-border investment in less developed economies will face greater opportunity cost than investment in developed economies. Second, according

to the theory of irreversible investment, multinational investors may face higher sunk costs in less-developed economies when they are hit by adverse shocks. Third, developed economies have greater geographical advantages than less-developed economies in terms of relevant laws and regulations, infrastructure such as transportation and communication and residents' purchasing power.

## 5. Discussion and Conclusions

### 5.1. Discussion

The research results of this paper can provide certain reference value for policy makers, mainly including the following points.

From the perspective of international relations, maintaining world peace and social stability and building good national relations are the most effective ways to defuse the negative impact of geopolitical risks on economic activities. Therefore, a stable political and economic environment is essential to attract foreign investment, encourage the development of foreign trade and thus promote their own economic growth. Countries should strengthen political and economic cooperation, deepen friendly exchanges, jointly build a community of shared future for humankind and promote the common development of the global economy. Countries should effectively defuse geopolitical risks through proactive diplomatic strategies, promote the signing of bilateral investment agreements, promote the establishment of transnational credit investigation systems, improve relevant laws and regulations and provide a sound economic and political environment and legal guarantee for international investment.

From a trade dependence standpoint, countries that rely on trade may have more perfect trade and investment protection mechanisms to maintain their own economic development, and they are more willing to sign relevant investment agreements with other countries to attract foreign investment. Hence, no single country can be fully equipped to deal with the growing social, economic and environmental risks facing the world. In this geopolitical context, major powers need to take a leadership role to enhance global resilience through innovative cooperation. Countries should reduce trade barriers and reject trade protectionism, thus reducing the possibility of political and economic instability and violent conflicts. Increased economic interdependence among countries and free trade in goods and services will promote globalization and thus moderate the adverse effects of geopolitical risks.

From an investment motivation perspective, larger market size, more resource endowments and higher scientific and technological levels can improve the expected returns of foreign investors. As long as it is greater than the expected cost of increased geopolitical risks, it can weaken this negative impact. Consequently, a higher level of economic development can attract more international investment inflows, thus promoting domestic economic development again. This is a virtuous cycle. In addition, the natural resources and strategic resources owned by the country are also important factors to promote the inflow of foreign capital and promote the virtuous cycle. Countries should pay attention to the sustainable development of natural resources and encourage the invention and progress of advanced science and technology, so as to enhance the capacity of their economies to cope with risks and provide a guarantee for steady economic growth.

From the enterprise level, geopolitical fragmentation may create different competitive environments in different countries, and the COVID-19 pandemic may exacerbate this trend. Therefore, multinational enterprises need to enhance their own capacity to deal with various risks and jointly take risk-prevention measures with national governments.

Based on the abovementioned argument, countries with different levels of development should work together to provide a secure and sustainable international environment for world economic activities and achieve mutual benefit and win-win results. Developed countries should maintain their regional advantages and attract foreign investment through high-tech development and relatively perfect infrastructure. Furthermore, they should shoulder the responsibility of big countries, safeguard the multilateral trading

system, actively participate in the reform of the World Trade Organization and promote the multilateral and bilateral regional investment and trade cooperation mechanism. Emerging economies should strive to improve their own scientific and technological level and innovation capacity, maintain their own resource endowment advantages and promote infrastructure construction in a coordinated manner. Moreover, they should actively participate in regional economic cooperation, promote trade and investment liberalization and facilitation, expand investment space and improve the investment guarantee mechanism.

*5.2. Conclusions*

This paper selects the FDI flow data of 41 sample countries and regions from 2003 to 2020, uses the geopolitical risk index constructed by Caldara and Iacoviello as the core explanatory variable, and introduces the market seeking motive, natural resource seeking motive and strategic resource seeking motive as the control variable to study the influence of geopolitical risk of the host country on FDI inflow from the perspective of the investment motivation of transnational investors. Due to the individual differences among countries and regions and the differences in the time of geopolitical events, this paper adopts the cluster fixed-effect model and introduces the individual fixed effect and the time fixed effect. The results are as follows.

First, the benchmark regression results show that the geopolitical risk of the host country can significantly inhibit FDI inflow, and the market seeking motivation, natural resource seeking motivation and strategic resource seeking motivation are all important factors driving FDI inflow. Second, this paper changed the sample size, changed the index selection of the core explanatory variable and control variable, and conducted the clustering fixed-effect regression as the robustness test again, and the regression results were consistent with those obtained above. Third, this paper used the interaction term of geopolitical risk index and trade dependency as the core explanatory variable to carry out fixed effect regression again. The research finds that trade dependency has a moderating effect on the negative impact of geopolitical risk; that is, for countries or regions with high trade dependency, geopolitical risk will have less impact on foreign direct investment. Fourthly, through empirical studies, this paper finds that geopolitical risks in economies with different levels of economic development have different impacts. Geopolitical risks in emerging economies can significantly reduce the inflow of foreign direct investment, while their impacts on developed economies are not significant.

According to the existing research conclusions, geopolitical risks have a negative impact on a variety of economic activities in the world, especially those of new economies. Actively participating in the process of economic globalization will help reduce the impact of geopolitical frictions. Most of the current studies focus on exploring the impact of geopolitical risks, while the research on how to reduce or eliminate this negative impact is relatively lacking. This paper believes that the future research direction can be expanded from the following aspects: First, how to eliminate geopolitical frictions through international cooperation, and the regulatory mechanism analysis of the impact of regional economic and trade organizations on geopolitical risks. Secondly, further study the impact of interactions between economic factors and geopolitical risk. Analyze the effectiveness of different types of policies from multiple perspectives. Third, analyze the influence of geopolitical risk on transnational investors from the perspective of enterprises at the micro level.

**Author Contributions:** Conceptualization, M.Y. and N.W.; methodology, N.W.; software, N.W.; validation, M.Y. and N.W.; formal analysis, M.Y.; investigation, N.W.; resources, N.W.; data curation, N.W.; writing—original draft preparation, M.Y.; writing—review and editing, M.Y.; visualization, N.W.; supervision, M.Y.; project administration, M.Y.; funding acquisition, M.Y. All authors have read and agreed to the published version of the manuscript.

**Funding:** This research was funded by the Chinese National Social Science Fund Project, "Research on the Mechanism of the Impact of Trade Costs on the Quality of China's Export Products under the Background of Anti-globalization and the Improvement Strategy" (Grant No. 19BJL115).

**Institutional Review Board Statement:** Not applicable.

**Informed Consent Statement:** Not applicable.

**Data Availability Statement:** The data that supports the fundings of this paper are available upon reasonable request from the corresponding author.

**Conflicts of Interest:** The authors declare no conflict of interest.

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
