# Peer review of "The Influence of Geopolitical Risk on International Direct Investment and Its Countermeasures"

_sustainability, doi:10.3390/su15032522_

Round 1
Reviewer 1 Report
Dear Authors,
Thank you for the opportunity of reading your manuscript. Attached you may find my review.
Best of luck in your research. All the best.

Author Response
Dear expert reviewer,
Thank you very much for your criticism and guidance. Please kindly check the attachment. We have carefully revised the manuscript by incorporating all the suggestions by the review panel. We hope this revised manuscript has addressed your concerns and look forward to hearing from you.
Sincerely,
The Authors

Reviewer 2 Report
1. Introduction to the article - short but well researched. Numerous references to the literature have been cited.
2. Theoretical Basis and Research Assumptions have been developed, but the article lacks a critical review of the literature on the topic.
3. Line 123 of the article - a mathematical expression is quoted, no numbering of expressions in the text, Authors should explain all expressions in the dependency, e.g.
- ?0 - ....................;
- ?1 - .......................; e.t.c.
4. The same remark applies to dependency 173 article line number (note as above).
5. No space in Fig. 1 and the text of the article - should be corrected.
6. In the opinion of the reviewer, the article contains too short subchapters, note to the Authors of the article: please combine the selected subchapters with the main chapter.
Author Response

(The authors gave the same response as above.)

Reviewer 3 Report
Dear Authors,
Greetings,
Please address all my comments before potential acceptance. See attached review report. Best Regards

Author Response

(The authors gave the same response as above.)

Reviewer 4 Report
The article is very interesting and deals with a topic of great interest, namely the influence that geopolitical risk has on foreign direct investments.
However, we believe that the article can be improved by the following elements:
1. A discussion section should be introduced so that the research results are related to other results previously obtained and published in articles in the specialized literature.
2. Tailoring policy recommendations to certain types of countries or including arguments to show why they are universally valid.

Author Response

(The authors gave the same response as above.)

Reviewer 5 Report
This study is a powerful work that uses a fixed-effects model to examine how such international tensions affect direct investment in light of the frequent geopolitical conflicts and global economic recession in recent years.
The analysis is precise and the authors' efforts are evident here and there. I am grateful for the opportunity to read such an interesting study.
The range of data examined by the authors covers 41 countries between 2003 and 2020, from which the research results are compelling in their exploration of the relationship between geopolitical risk and FDI flows.
However, the results from this study, according to the authors, boil down to the all too obvious fact that geopolitical risk can significantly impede FDI inflows and discourage foreign direct investment, and unfortunately, the originality and novel contributions of the study are not strong enough to be considered. Even so, however, this research has the potential to become a cornerstone of future research.
In light of the unpredictable trends in global economic and geopolitical tensions, I propose that this study focus on the aspects of confirming the validity of the analytical models and measures and providing effective guidelines for further research activities, rather than appealing to the reader with the results obtained.
1. Given that factors such as market size, natural resources, and science and technology are important factors in attracting FDI, and further emphasizing the aspect of trade dependence that has the effect of mitigating the negative impact of geopolitical risks on FDI, and the new important element, it should describe how the possibility of eliminating geopolitical frictions through economic cooperation should be explicitly incorporated into the analytical model.
2. We would love to hear the authors' discussion further of the policy implications of the less significant impact of geopolitical risks on outward FDI in developed countries. Perhaps you could have extended more the section of Recommendation as well.
Please note that all items specified in the MDPI should be covered at the end of the text. The draft only mentions external funding, but please cover other factors as well. Good luck!
Author Response

(The authors gave the same response as above.)

Round 2
Reviewer 1 Report
Dear Authors,
Thank you for the corrections you have made to the manuscript. You have improved its quality.
Attached you may find my revision. Good luck. Best regards.

Author Response
Dear expert reviewer,
Thank you very much for your criticism and guidance. Please kindly check the attachment. We have carefully revised the manuscript by incorporating all the suggestions by the review panel. We hope this revised manuscript has addressed your concerns and look forward to hearing from you.
Best regards and Happy Chinese New Year!
The Authors

Reviewer 3 Report
Comments
Authors addressed my comments. However, The paper needs a logical flow with good editing. Please link all parts together.
Author Response
Dear expert reviewer,
We are very grateful and honored to receive your guidance and suggestions. We have re-edited the paper format and citation style according to the format required by the journal. We hope you will find this revised version satisfactory. If you have more suggestions or questions, please kindly let me know.
Best regards and Happy Chinese New Year!
The Authors
Round 3
Reviewer 1 Report
Dear Authors,
Thank you for resubmitting the paper and for taking my suggestions into consideration.
I think your manuscript has significantly improved. You are now citing the literature as requested by the journal and you have added the AIC and BIC goodness-of-fit measures to the tables. Even though I still think the R2 statistic being too low is a shortcoming of your paper because it indicates a lack of explanatory power of the model, I understand your comments on the current state of the art of the research in this topic and accept your methodology as it is. However, for the future, in my (very) humble opinion I think you should look into more advanced and robust models. For me, estimating a macroeconomic variable with a linear regression is a very simplistic approach.
I am now convinced that there are no multicollinearity issues, based on your VIF values. Maybe, if you think it is relevant, you can add them to your paper to show to the reader that there are no issues in your regression. I would encourage you do so, but I leave that decision up to you.
Overall, I am now convinced this manuscript has improved sufficiently to be accepted for publication. Thank you for your patience and commitment in answering my comments; I think they have significantly improved your paper's quality.
Congratulations, and best of luck in your future research endeavours!